# Fabrication and Microstructure of Laminated HAP–45S5 Bioglass Ceramics by Spark Plasma Sintering

**DOI:** 10.3390/ma12030484

**Published:** 2019-02-04

**Authors:** Ye Meng, Wenjiang Qiang, Jingqin Pang

**Affiliations:** 1School of Materials Science and Engineering, University of Science and Technology Beijing, Beijing 100083, China; wjqiang@mater.ustb.edu.cn (W.Q.); jqpang2010@163.com (J.P.); 2National Demonstration Center for Experimental Materials Education, University of Science and Technology Beijing, Beijing 100083, China

**Keywords:** hydroxyapatite, 45S5 bioglass, laminated, spark plasma sintering, interfaces structures

## Abstract

Hydroxyapatite (HAP) has excellent biocompatibility with living bone tissue and does not cause defensive body reactions, therefore, it has become one of the most widely used calcium phosphate materials in dental and medical fields. However, its poor mechanical properties have been a substantial challenge in the application of HAP for the replacement of load-bearing or large bone defects. Laminated HAP–45S5 bioglass ceramics composites were prepared by the spark plasma sintering (SPS) technique. The interface structures between the HAP and 45S5 bioglass layers and the mechanical properties of the laminated composites were investigated. It was demonstrated that there was mutual transfer and exchange of Ca and Na atoms at the interface between 45S5 bioglass/HAP laminated layers, which contributed considerably to the interfacial bonding. Due from the laminated structure and strong interface bonding, laminated HAP–45S5 bioglass is recommended for structural applications.

## 1. Introduction

In recent years, laminated ceramics have attracted considerable attention due to their high fracture toughness, fracture work, and flexural strength [1,2,3,4,5]. In the 1990s, based on the bionic structures, such as bamboo and shells, Clegg designed and prepared SiC/graphite laminated composite ceramics, which improved the fracture toughness of laminated composite ceramics from 3.6 MPa·m^1/2^ to 15 MPa·m^1/2^ and the fracture work from 28 J·m^−2^ to 4625 J·m^−2^ [6]. Since then, the research on laminated composites has been unprecedentedly active. Xie et al. [7] prepared laminated SiC_w_/SiC ceramic composites by the combination of chemical vapor infiltration and tape casting. The volume fraction of whiskers reached as high as 40 vol.%, and the flexural strength, tensile strength, and fracture toughness were 315 MPa, 158 MPa, and 8.02 MPa·m^1/2^, respectively. Since then, more laminated ceramic materials have been developed, including laminated SiC ceramics [8,9,10] and laminated Si_3_N_4_ ceramics [11,12].

Hydroxyapatite (HAP) possesses good bioactivity and compatibility with roughly the same component and crystal structure as the human skeleton [13]. However, the application of HAP has been largely limited by its low fracture toughness (0.8–1.2 MPa·m^1/2^) and flexure strength (140 MPa) [14]. To improve the mechanical properties of HAP, reinforcements such as zirconia, alumina, carbon fiber, and carbon nanotubes are incorporated to make HAP-based composites [15,16,17,18]. 

45S5 bioglass is a biocompatible material with remarkable osteoconductivity, osteoinductivity, and controllable biodegradability [19,20,21,22]. Hench et al. developed 45S5 bioglass, composed of 45 wt.% SiO_2_, 24.5 wt.% Na_2_O, 24.5 wt.% CaO, and 6 wt.% P_2_O_5_ [23]. The 45S5 bioglass can be further strengthened by the incorporation of metal, polymer, and fiber [24,25,26].

Recently, many attempts to combine HAP with bioactive glasses have been reported, such as sol-gel [27,28,29,30], hot-press sintering [31,32], pressureless sintering [33,34], and SPS [35,36,37]. HAP composite produced with sol-gel has been mainly used as a porous coating on metallic implants in orthopedic and dental applications to combine the excellent mechanical properties of metals and bioactivity of hydroxyapatite [28]. The higher temperature (1200–1350 °C) required for the conventional sintering (hot-press sintering, pressureless sintering) of this composite system results in the complete crystallization of bioglass and excessive reactions between bioglass and HAP, which are suspected to delay the bioactivity response [35]. SPS is a pressure-assisted synthesis and processing technique which employs high current (pulsed direct current) and low voltage. In the SPS technique, there is direct heating of the sample in a graphite mold. A pulsed direct current (DC) passes through graphite punch rods and dies simultaneously with uniaxial pressure [38]. The SPS method allows fine tuning of the sintering process (pressure, temperature, time, and higher heating rates) which is crucial for HAP-bioglass sintering. In comparison with conventional sintering techniques, higher heating rates (between 100 and 600 °C/min) can be achieved. With such high heating rates, densification mechanisms are favored [39]. Grasso et al. [40]. reported the sintering of bioglass powder by SPS with temperature (350–500 °C) and pressure (70–300 MPa), respectively. They found that the density of the samples is strongly affected by the applied pressure. When the sintering temperature was below Tg, high pressure promoted the compaction of powders. In order to achieve samples with density higher than 95%, a sintering pressure of 300 MPa was needed at 500 °C. By increasing the sintering temperature up to 550 °C, 70 MPa of pressure was enough to achieve samples with a density exceeding 98% [40]. Gergely et al. [41]. confirmed that due to the rapid sintering time (5 min) applied during the SPS process, the phase composition was not changed with the temperatures (800–950 °C) used. Dubey et al. [42]. observed that HAP began to dissociate to β-tricalcium phosphate (β-TCP) when SPS was sintered at 1200 °C. As has been reported, composites of HAP and TCP induce moderate initial inflammatory responses compared with HAP alone [43]. β-TCP, HAP, and processed spongiosa are common materials for filling bone defects. Bone grafts become incorporated by the host bone and are substituted either completely (β-TCP, processed spongiosa) or partially (HAP) [44]. Therefore, we hope to sinter HAP-bioglass laminated composites without decomposition of HAP into β-TCP. SPS requires relatively low temperatures to reach high consolidation levels. Consequently, SPS is beneficial in the sintering processes where crystallization phenomena, grain growth, and/or phases decomposition have to be avoided or minimized [36]. There are two main motivations for sintering calcium phosphates with a glassy phase: on one hand, it is possible to tune the dissolution of the final system to enhance its biological response through the synergistic combination of two bioactive phases; on the other hand, the glass acts as a sintering aid with the aim to increase the densification of the composite and thus its mechanical strength [45]. 45S5 bioglass-coated implants exhibited greater bone ingrowth compared to hydroxyapatite (HA)-coated and as controls (CTL) implants, and they maintained their mechanical integrity. The bioactive glasses are commonly more reactive than HA, and hence the combination of HA and bioactive glasses is promising and may lead to the development of new-generation composites with tailored biological properties [46]. Several studies [47,48,49,50] have described metalo-ceramic composites obtained by fine tuning of SPS sintering properties. These studies all use intermetallic compounds as reinforcing phases to improve the overall hardness, strength, or wear resistance of the composite structure, while tough metal phases provide appropriate ductility, fracture toughness, and machinability. The inherent SPS technology features, such as rapid heating rates, short sintering times, and applied external pressure (which improves the densification) [51], enable excellent sintering while avoiding formation of microstructural issues related to, e.g., excessive grain growth, development of porosity, or microstructural heterogeneities.

In this study, laminated structures were introduced into structural materials. Laminated HAP–45S5 bioglass ceramics composites were prepared by SPS. In addition, monolithic HAP ceramics were fabricated under the same conditions for comparison. The microstructure evolution and mechanical properties were thoroughly investigated.

## 2. Materials and Methods 

HAP powder (molecular formula is Ca_5_HO_13_P_3_) with purity of 97% and d_90_ < 200 nm was provided by Sigma-Aldrich. Co., St. Louis, MO, USA. The 45S5 bioglass powder was provided by MO-SCI Corporation in the Rolla, MO, USA, with a chemical composition (wt.%) of 45% SiO_2_, 24.5% Na_2_O, 24.5% CaO, and 6% P_2_O_5_ and d_90_ < 6 μm. The mass ratio of HAP:45S5 bioglass was 2:1. Ten g HAP or 45S5 bioglass were mixed in 60 mL alcohol by magnetic stirring for 2 h at the speed of 300 rpm, and then dried in an oven at 343 K for 15 h. 

The powders of the corresponding quality are put into the graphite die (Beijing Sanye Carbon Co., Ltd., Beijing, China) layer by layer in a certain order (Figure 1a). The masses of the HAP layer and the 45S5 bioglass layer were 1 g and 0.5 g, respectively. The inner diameter of the graphite mould is 20 mm. And a sheet of graphite paper was placed on the inner wall of the mould to facilitate the demoulding. Each layer was loaded axially with a pressure of 20 MPa to fabricate the preform. The preform was consolidated by SPS equipment (SPS. Dr Sinter 1050, Sumitomo Coal Mining Co., Ltd., Tokyo, Japan) at 1223 K, 40 MPa (process planning according to Ref. [40,52,53]). The heating rate was increased from room temperature to 1123 K at 100 K/min, then decreased to 50 K/min to 1173 K, 30 K/min to 1203 K, 20 K/min to 1223 K, and kept for 5 min at 1223 K. The intent keeps the temperature stable during the holding process on the basis of rapid sintering. At the initial stage of sintering, 10 MPa pressure was applied on the head, and 10 MPa pressure was increased at 873 K, 973 K, and 1073 K, respectively, then 40 MPa pressure was maintained at thermal insulation (Figure 1b). Afterwards, the pressure was reduced slowly to 0 MPa before the temperature drops to 873 K to release the internal stress sufficiently during the cooling process and to avoid the cracking of the sample.

Finally, the samples were cut into bars with dimensions 2.5 mm × 2.5 mm × 18 mm to examine the mechanical properties and microstructure. The microstructures and fractured surfaces were characterized by field emission scanning electron microscopy (FESEM, Zeiss Supra55, Oberkochen, Germany). The 3-point bending test is measured by the Microcomputer Control Universal Test Machine (WDW-1, MTS Systems Corporation, Eden Prairie, MN USA) with a loading rate of 0.5 mm/min. 

## 3. Results and Discussions

Figure 2 shows the SEM images of the raw materials of HAP and 45S5 bioglass. It is shown in Figure 2a that the raw HAP powder is spherical with a relatively uniform particle size and the average size as 200 nm. In contrast, as shown in Figure 2b, the 45S5 bioglass powder particles are highly irregular. The large particles (~5 μm) with sharp edges are attached massive small particles (1 μm).

Figure 3 shows the SEM image of the fresh fracture surface of the laminated HAP–45S5 bioglass ceramic composite. As clear, the HAP–45S5 bioglass composite, in which the 800 μm thick 45S5 bioglass is favorably sandwiched by grey HAP layers, shows a well-laminated structure with clear and parallel interfaces. Noteworthy that such composite is sintered by SPS at a low temperature (1223 K) and short time (16 min), which are superior to traditional sintering techniques such as hot pressing which generally requires a higher temperature of 1273–1573 K and longer sintering time of 60–120 min [54,55,56,57,58]. The rapid sintering can effectively suppress the grain growth and excessive interfacial reaction, which are favorable for the mechanical properties of the composites.

Figure 4 shows the SEM images of the fracture surface of laminated HAP–45S5 bioglass ceramics composite. As seen, the 45S5 bioglass layer presents a porous microstructure (Figure 4b) with isolated, irregularly shaped pores surrounded by dense walls. The HAP layer displays a stratified distribution of two kinds of grains. The HAP particles near the side of 45S5 bioglass layer are larger (Figure 6c) with an average particle size of 2 μm, while the HAP particles far from the 45S5 bioglass layer are smaller (Figure 6d) with an average particle size of 1 μm. Figure 4b–d shows the characteristics of the three layers. In the 45S5 bioglass layer, holes of different sizes are distributed in the layer. From the hole, it can be seen that the 45S5 bioglass maintains its original shape and the particle size is basically kept within 1–2 μm. This porosity could be formed by the development of the gaseous byproducts during the sintering (Figure 4b) [21]. The grains of the HAP close to the 45S5 bioglass layer grow obviously, up to 2 μm. It can be seen the melting phenomenon of 45S5 bioglass in contact with HAP. On the contrary, the 45S5 bioglass far from HAP retains its original shape and pore structure. This indicates that the sintering temperature at the contact point of 45S5 bioglass with HAP is higher than those in the phase of 45S5 bioglass or HAP. The relatively high temperature melts 45S5 bioglass and causes the grain growth of HAP (Figure 4c). Hence, the addition of glass can significantly enhance the sinterability of HAP [59]. As a result, the HAP grains far from 45S5 bioglass retain their fine and compact structure with a particle size of 1 µm (Figure 4d). Meanwhile, it can be seen that the samples possessed a rather compact structure with good interfacial bonding. Compared with porous materials, the bacterial counts of *Staphylococcus* growing on the surface of smooth materials is lower [44]. Staphylococci are considered the most infecting microorganisms of bone tissue, with *Staphylococcus aureus* being the most commonly found species. The composites have advantages in avoiding infection and complications when it is used as an intraosseous implant to repair alveolar ridge after tooth loss [60].

To reveal the diffusion phenomena at the interface, the interface of the laminated HAP–45S5 bioglass ceramics was analyzed by Energy dispersive X-ray spectroscopy (EDX), as shown in Figure 5. From the 45S5 bioglass layer to the HAP layers, the amount of Si gradually decreases. At the same time, the amount of P gradually reduces from the HAP layer to the 45S5 bioglass layer. In addition, in the portion of coarse grain HAP adjacent to 45S5 bioglass, Ca content is decreased while Na content is increased. The structure of HAP can easily accommodate a great variety of anionic and cationic substitutes [61]. In this layer, Ca ions are replaced by Na ions [62], and thus the opposite diffusion path of Ca and Na occurs. The diffusion of Na out from the glassy phase and its replacement from the diffusion of Ca from the CaP based phase makes the BG less vulnerable towards crystallization [63].

To clearly observe the migration and change of the main elements of the HAP/45S5 composites before and after sintering, the atomic percentages of the elements (Si, Na, Ca, and P) in 45S5 bioglass and HAP are based on the EDX spot results, which are listed in Table 1.

As seen, the contents of Si and P in the 45S5 bioglass layer do not change significantly. At the same time, the Na concentration decreases by nearly a factor of 2/3, while the Ca concentration is increased by more than a factor of 2. It can be seen that the Na and Ca atoms in the 45S5 bioglass layer are transferred in the opposite directions in the composites, as shown in Figure 6a. A very small amount of Si appears in the HAP coarse grain layer adjacent to the 45S5 bioglass layer, which transfers from 45S5 bioglass to HAP during sintering. Meanwhile, in the HAP coarse grain layer, the P content decreases slightly, while the Ca content decreases by nearly 60%, and the Na content reaches 52.44%. It can be seen that in the HAP coarse grain layer, Na and Ca atoms also transfer in the opposite direction, and this phenomenon is contrary to that in the 45S5 bioglass layer. Therefore, it can be concluded that the Ca and Na atoms exchange between 45S5 bioglass layer and HAP coarse grain layer. Meanwhile, it can be seen that a very small amount of P atoms was transferred from the HAP layer to the 45S5 bioglass layer, as shown in Figure 6b. The trend of decrease in Ca content and increase in Na content in the HAP fine grain layer tends to decrease, and the percentage of P atomic mass remains almost unchanged. No evidence of Si atoms can be found. It can be seen that the far distance from the 45S5 bioglass layer to HAP lead to a long time for atom transfer due to the increase of diffusion activation energy, as shown in Figure 6c. The P atoms and Si atoms are maintained in their original positions and do not diffuse, as elements undergoing diffusion can bond closely with each other. Also, the presence of C atoms is detected in all three layers after sintering. It is well-accepted that carbonates are easily generated in Ca-containing minerals. Besides, the mechanism of carbonation depends on the type of formed carbonates and the type of substrates [64]. The existence of large amounts of carbon in the graphite mold makes it easy for C atoms to enter the bioglass interior. 

Five repeated three-point bending tests were carried out for HAP–45S5 bioglass laminar composites and monolayer HAP composites, and the bending strength was 28 ± 10 MPa and 79.8 ± 14 MPa, respectively. The stress–strain curves are drawn from the three-point bending data of two groups of tests whose bending strength is closest to the average value. Figure 7 shows the engineering stress–strain curves of monolayer HAP composite and HAP–45S5 bioglass laminar composite. At the beginning of the strain, the stress of the monolayer HAP composite shows a significant increase and the fracture occurs when the strain reaches 0.31% with a corresponding stress of 42.2 MPa. During the initial period, the strain of the HAP–45S5 bioglass layered composite reaches 1% and the stress increases from 2.1 MPa to 3.8 MPa. At the strain of 1%, the stress increases significantly with the increase of strain. When the strain reaches 5.9%, the stress achieves the maximum value of 79.8 MPa. The strain and stress of HAP ceramics both are increased by 19 times and 1.89 times after adding 45S5 bioglass as the intermediate layer. Therefore, the lamellar structure is very beneficial for the improvement of the toughness of the material. The results indicated that HAP–45S5 bioglass laminar composites have significantly higher strength compared to HAP-bioglass monolayer sintered bodies reported previously [65].

Previously discussed by several authors in the literature [27,43], HAP-Bioglass composites do not negatively affect the cells. The same holds both for fibroblasts and for osteocytes [66,67]. Compared to the conventional sintering practice [31,32,33], the milder processing conditions during the SPS process, such as the compaction pressure, sintering time, and temperature, produced HAP–45S5 bioglass laminar composite without excessive reactions between the constituents and prevented the crystallization of the bioglass [35]. Compared with autogenous bone transplantation, synthetic biomaterials are available indefinitely, relatively inexpensive, completely sterile, and require no additional surgery [43,68]. Many studies also show that synthetic bone transplantation can effectively respond to soft tissue growth, new bone formation, and biodegradation after transplantation [69]. Results indicated that improvement of the mechanical properties could be obtained with the stratified structure, and suggest that HAP–45S5 bioglass laminated composite might be a good candidate for biomedical applications such as scaffold materials [35,53,70], bone grafts used in dentistry [37,41,43,71], filling bone defects [33,36,44], and facial bone reconstruction [38,69] for bone regeneration/repair. 

## 4. Conclusions

Laminated HAP–45S5 bioglass ceramics were successfully prepared by SPS, in which the HAP layers and 45S5 bioglass layers are alternately attacked with smooth and strong interfacial bonding. Remarkable diffusion of Ca and Na takes place at the interface between the 45S5 bioglass layer and the HAP coarse grain layer. Meanwhile, a moderate P diffusion also occurs from the HAP layer to the 45S5 bioglass layer. Compared to HAP ceramics, the laminated HAP–45S5 composite showed a significant increase of tensile strength (19 times) and elongation (1.89 times). Therefore, the incorporation of 45S5 bioglass to create the laminated composite structure is highly favorable for greatly enhancing the mechanical properties of HAP ceramics.

## Figures and Tables

**Figure 1 materials-12-00484-f001:**
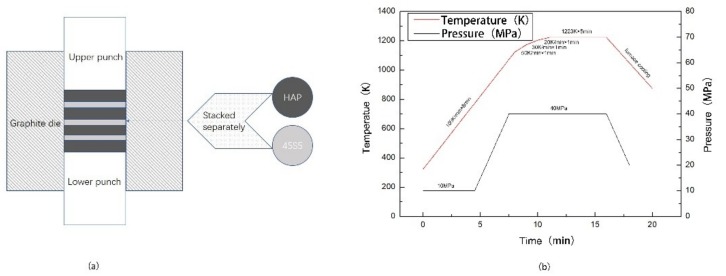
Preparation of laminated HAP/45S5 bioglass ceramic: (**a**) The green body structure of HAP–45S5 bioglass laminated composite material; (**b**) The sintering process by spark plasma sintering.

**Figure 2 materials-12-00484-f002:**
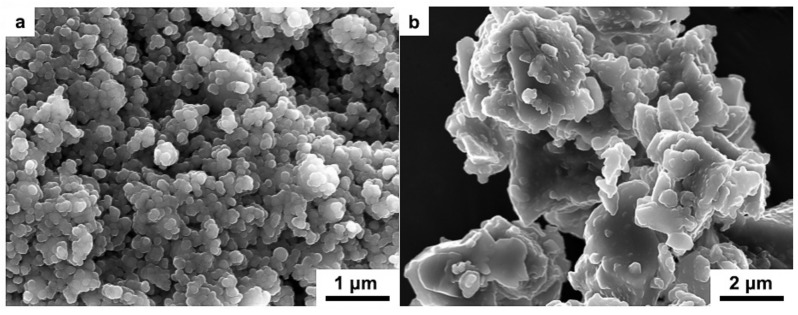
SEM images of raw materials: (**a**) pure HAP powder and (**b**) pure 45S5 bioglass powder.

**Figure 3 materials-12-00484-f003:**
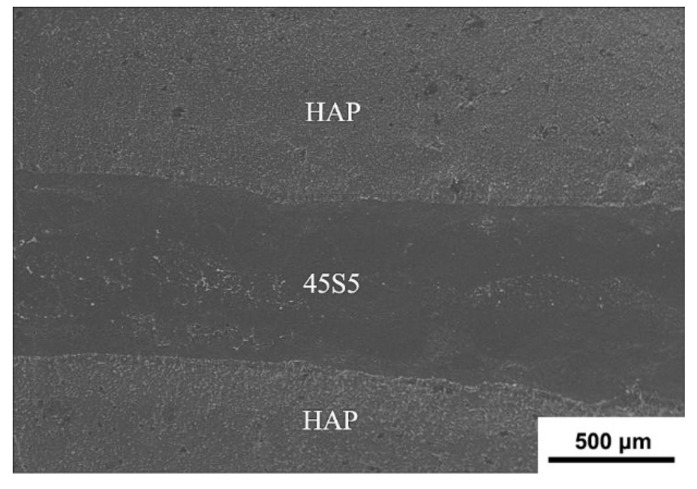
SEM micrograph of the fracture surface of laminated HAP–45S5 bioglass ceramics.

**Figure 4 materials-12-00484-f004:**
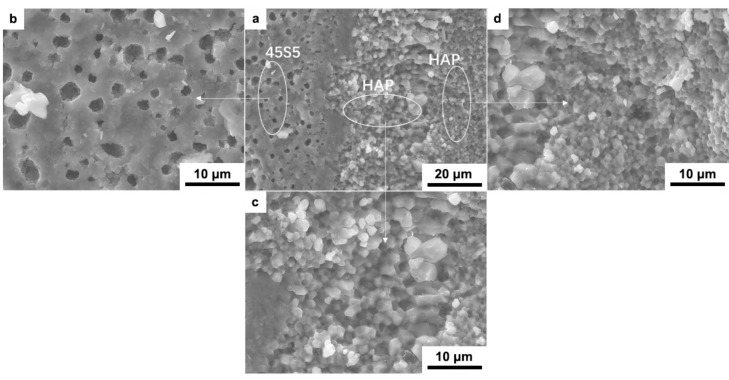
SEM images of the fracture surface of laminated HAP–45S5 bioglass ceramics (**a**); enlarged figure of 45S5 bioglass layer (**b**) and HAP layers (**c**,**d**).

**Figure 5 materials-12-00484-f005:**
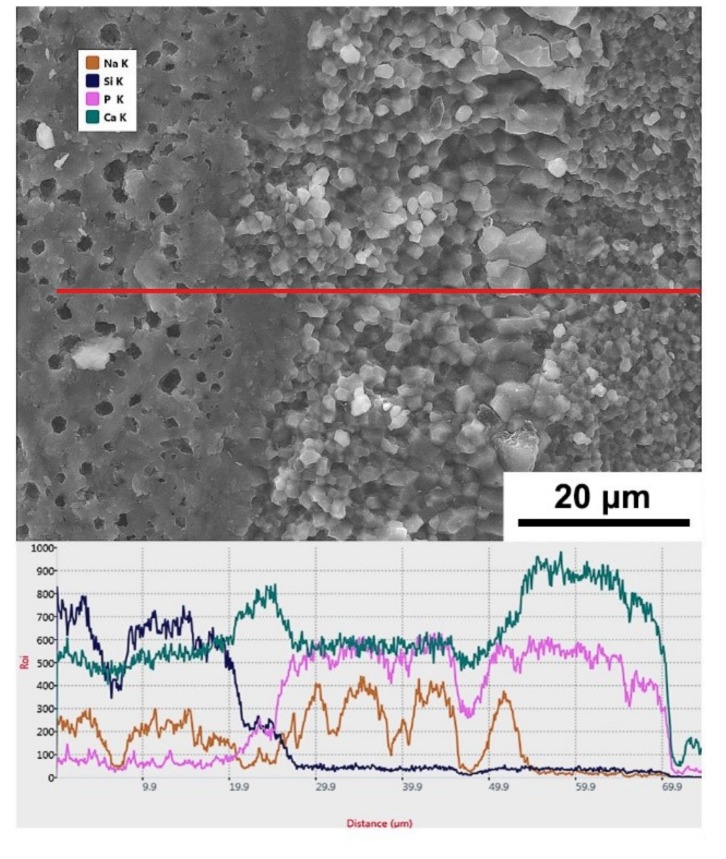
SEM micrograph of laminated ceramics fracture surface and element diffusion analysis of EDX.

**Figure 6 materials-12-00484-f006:**
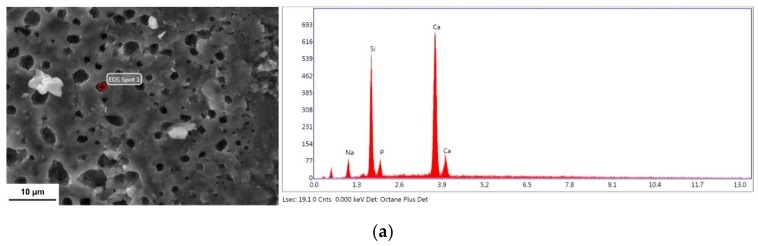
EDS patterns of the fracture surface of HAP–45S5 bioglass composite: (**a**) 45S5 bioglass layer; (**b**) HAP coarse grain layer and (**c**) HAP fine grain layer.

**Figure 7 materials-12-00484-f007:**
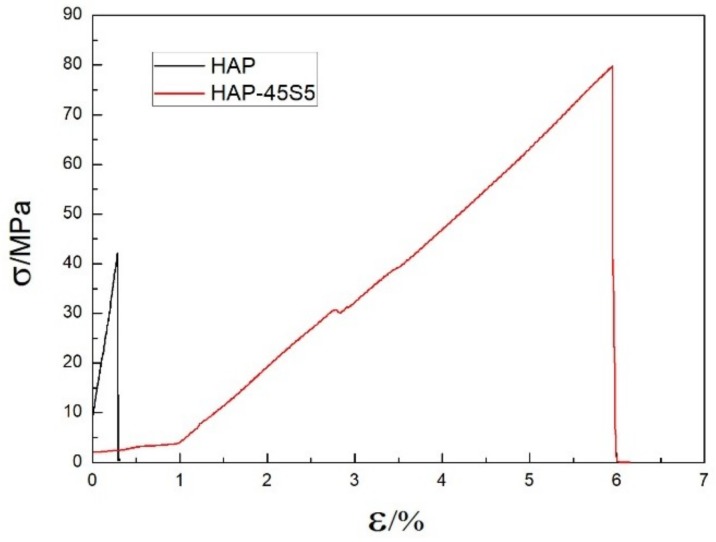
The engineering stress–strain curves of monolayer HAP composite and HAP–45S5 bioglass laminar composite.

**Table 1 materials-12-00484-t001:** Atomic weight percentage of Si, Na, Ca, and P in raw powder of 45S5 bioglass and HAP and those in laminated HAP–45S5 bioglass ceramics.

Sample		Atomic Weight Percentage/%			
		Si	Na	Ca	P
Raw powder	45S5 bioglass	35.31	30.93	29.59	4.17
	HAP	0	0	68.26	31.74
Sintered specimen	45S5 bioglass layer	36.52	11.69	71.69	6.21
	coarse grain HAP layer	0.97	52.44	24.00	22.60
	fine grain HAP layer	0	7.21	60.5	32.29

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
