# Peer review of "Fabrication and Microstructure of Laminated HAP–45S5 Bioglass Ceramics by Spark Plasma Sintering"

_materials, 2019, doi:10.3390/ma12030484_

Round 1
Reviewer 1 Report
Thanks to authors for great, and well-planned research did.
All paper parts: abstract, materials and methods, result, and conclusions well written and clearly presented.
The paper in common is well-organised and cohesive.
I would like to suggest authors enhance the introduction, by the giving small (2-3 sentences) off common approaches used for HAP and bioglass modification
To the ALL PAAPER please check spaces:
BEFORE the [ref], - it should be (lines 31, 35, 38, 43, 45, 50, 54, ….
if a unit is abbreviated as one character, there must not be a space between the number and the unit (e.g., 5m, 26K).
If the unit is abbreviated as two or more Characters, there must be a space between the number and the unit (e.g., "0.8-1.2 Mpa", "10g").
Line 32: What is SiCw - ?
Line 59: “average size of less than 200nm” should be changed to “ d50 or d90 <200 nm”
Line 62: see the previous comment
Line 62: “ The powders … die layer by layer…” layer thickness (approx) or each layer portion in g.
Lines 670-71: unclear “then decreased to 50K/min, 30 K/min, 20K/min” what time for each period were used?
Lines 72-75: please provide Pressure/Temperature diagram, instead of currently presented fig1. Currently, presented fig1 is too obvious.
TO Fig 5. If it possible withdraw colour curves from the b/w SEM image BUT put tightly below or above (as one image, not on to other pages), keeping start/end of the curves exactly as in SEM image. It will significantly improve image readability.
TO discussion part
Is it present study based on the ONE HAP and ONE HAP-45S5 specimens tests results?
For results, standard deviation and experiment repeatability should be tested at least 5 specimens. Standard deviation should present for the measurements.
I suggest improving discussion part by supporting of literature data of HAP, HAP/polymer composite,
Author Response
Dear Reviewer:
First of all, thank you for your recognition of our research work, which is a great encouragement and affirmation to us. We have carefully studied your valuable opinions on us and have revised and supplemented them one by one. The responds are as following;
Point 1: I would like to suggest authors enhance the introduction, by the giving small (2-3 sentences) off common approaches used for HAP and bioglass modification
Response 1: As suggested by reviewer, in the introduction, we added relevant contents about common methods of modification of HAP and bioglass, and gave a brief description of the characteristics of these methods. lines46-56.
Point 2: Before the [ref], - it should be (lines 31, 35, 38, 43, 45, 50, 54, ….
if a unit is abbreviated as one character, there must not be a space between the number and the unit (e.g., 5m, 26K).
If the unit is abbreviated as two or more Characters, there must be a space between the number and the unit (e.g., "0.8-1.2 Mpa", "10g").
Response 2: We are very sorry for our negligence of the problem of spaces between numbers and units. We have checked all the numbers and units in the article one by one and revised them in full accordance with the standard format provided by reviewer. Lines 30,34,35,39,43,58,84,96, ….
Point 3: Line 32: What is SiCw - ?
Response 3: Line 32: the “SiCw” comes from the paper “Fabrication of laminated SiCw/SiC ceramic composites by CVI” ( Xie, Y.; Cheng, L.; Li, L.; Mei, H.; Zhang, L. J EUR CERAM SOC 2013, 33, 1701-1706. ). The “w” means whiskers. The author prepared a slurry of SiC Whiskers by ball milling, then manufactured a sheet of SiC Whiskers by tape casting. Thirdly, a substrate of SiCw/SiC was prepared by CVI. Finally, a laminated SiCw/SiC ceramic was fabricated by alternately tape casting and CVI on the substrate. The purpose of "w" is to distinguish SiC whiskers from SiC matrix.
Point 4: Line 59: “average size of less than 200nm” should be changed to “d50 or d90 <200 nm”
Line 62: see the previous comment
Response 4: Line 59, 62: We are very sorry for our incorrect writing about the term for average grain size. The statement of average grain size in the original manuscript is based on the manufacturer's product description. After receiving comments from Reviewer, we measured the particle size and distribution of these two kinds of powders by laser diffraction particle size analyzer. We modified the particle size of the two powders to: HAP powder (d90<200 nm), 45S5 bioglass powder (d90<6 μm).
Point 5: Line 62: “The powders … die layer by layer…” layer thickness (approx) or each layer portion in g.
Response 5: Line 62: It is really true as Reviewer suggested that there is no exact value for the amount of each layer in the manufacture of layered materials. We describe the exact grams used for each layer of material. The mass of HAP layer and 45S5 bioglass layer was 1g and 0.5g respectively.
Point 6: Lines 70-71: unclear “then decreased to 50K/min, 30 K/min, 20K/min” what time for each period were used?
Response 6: Lines 70-71: We regret that the sintering process has not been clearly stated. We have revised the description of the process as follows. “The heating rate was increased from room temperature to 1123K at 100 K/min, then decreased to 50 K/min to 1173K, 30 K/min to 1203K, 20 K/min to 1223K, and kept for 5 minutes at 1223K.”
Point 7: Lines 72-75: please provide Pressure/Temperature diagram, instead of currently presented fig1. Currently, presented fig1 is too obvious.
Response 7: Lines 72-75: As Reviewer suggested, we added pressure/temperature diagrams to describe the sintering process more accurately and intuitively, named Fig1b.
Point 8: To Fig 5. If it possible withdraw colour curves from the b/w SEM image BUT put tightly below or above (as one image, not on to other pages), keeping start/end of the curves exactly as in SEM image. It will significantly improve image readability.
Response 8: Fig 5: Thank you very much for your suggestions on this map. we withdrew the color curves from the SEM image and put it below, kept start/end of the curves exactly as in SEM image. It does significantly improve the readability of the image and make it clearer and more professional.
Point 9: To discussion part
Is it present study based on the ONE HAP and ONE HAP-45S5 specimens tests results?
For results, standard deviation and experiment repeatability should be tested at least 5 specimens. Standard deviation should present for the measurements.
Response 9: To discussion part: Five repeated three-point bending tests were carried out for HAP-45S5 bioglass laminar composites and monolayer HAP composites, and the bending strength was 28±10 MPa and 79.8±14 MPa, respectively. The stress-strain curves are drawn from the three-point bending data of two groups of tests whose bending strength is closest to the average value. we have added the explanations to this part of the paper according to the Reviewer’s suggestion.
Point 10: I suggest improving discussion part by supporting of literature data of HAP, HAP/polymer composite.
Response 10: Considering the Reviewer’s suggestion, we have added several literature data of HAP, HAP/polymer composite to supporting our discussion. These supplementary literatures are in the microstructures, atom migration, mechanical properties and applications, respectively.
Thank you very much for your comments, which has greatly improved our paper from professional level to standard format.
Reviewer 2 Report
The manuscript is clearly written and the results are well presented. The topic of the manuscript falls into the Materials aim and scope.
The treated topic is interested and actual and therefore. The results appear to be valid.
It is not clear if The images under the microscope are without magnification measures, I suggest to better clarify them.
I also suggest that you enter the following paper in order to improve the discussion section with recent paper about possibile clinical application of the material treated. Some sample
Overall a good potential paper.
Cicciù M, Cervino G, Herford AS, et al. Facial Bone Reconstruction Using both Marine or Non-Marine Bone Substitutes: Evaluation of Current Outcomes in a Systematic Literature Review. Mar Drugs. 2018;16(1):27. Published 2018 Jan 13. doi:10.3390/md16010027
Author Response
Dear Reviewer:
First of all, thank you very much for your interest and recognition in our research work, which has greatly enhanced our confidence. We have carefully studied your valuable opinions on us and have revised and supplemented them one by one. The responds are as following:
Point 1: It is not clear if the images under the microscope are without magnification measures, I suggest to better clarify them.
Response 1: We are very sorry for our negligence of the unclear and inconsistent format of the magnification measures of the images. We have changed the rulers of all the images in the article into a uniform format and thickened them to enhance the contrast with the background. The revised images do seem clearer, and our articles as a whole look more beautiful and professional. Thank you very much for your suggestion.
Point 2: I also suggest that you enter the following paper in order to improve the discussion section with recent paper about possibile clinical application of the material treated. Some sample overall a good potential paper.
Cicciù M, Cervino G, Herford AS, et al. Facial Bone Reconstruction Using both Marine or Non-Marine Bone Substitutes: Evaluation of Current Outcomes in a Systematic Literature Review. Mar Drugs. 2018;16(1):27. Published 2018 Jan 13. doi:10.3390/md16010027
Response 2: We have carefully studied the review article you recommended. It is very helpful to enrich our paper on material therapy for possible clinical applications. we have added several literature data of applications clinical applications to supporting our discussion. These supplementary literatures are in the microstructures, atom migration, mechanical properties and applications, respectively.
Thank you very much for your comments, which has greatly improved our paper from professional level to standard format.
Round 2
Reviewer 1 Report
Dear Authors,
Thank You for paper improvement, everything is perfect,
I suggest to support SPS method selection by following text and reference (please feel free to modify English) Place - page 2 lines 46-56:
"The SPS method allows to fine tune of sintering process (pressure, temperature and time) [10.3390/met7010016] which is crucial for HAP bioglass sintering - reaching high temperature during a short period."
In paper [10.3390/met7010016] described metalo-ceramic composite obtaining by fine-tuning of SPS sintering properties.
Author Response
Dear Reviewer:
First of all, thank you for your recognition of our research work, which is a great encouragement and affirmation to us. We have carefully studied your valuable opinions on us and have revised and supplemented them one by one. Revised portion are marked in blue in the paper. The responds are as following;
Point 1: I suggest to support SPS method selection by following text and reference (please feel free to modify English) Place - page 2 lines 46-56:
"The SPS method allows to fine tune of sintering process (pressure, temperature and time) [10.3390/met7010016] which is crucial for HAP bioglass sintering - reaching high temperature during a short period."
Response 1: Considering the Reviewer’s suggestion, we have added 8 references on fine-tuning of SPS sintering process, in line 53-74. The contents include the influence of temperature, pressure, holding time and heating rate of SPS sintering on the sintered samples.
Point 2: In paper [10.3390/met7010016] described metalo-ceramic composite obtaining by fine-tuning of SPS sintering properties.
Response 2: As Reviewer suggested that we have added the advantage of SPS in sintering metal-ceramic composites, in line 84-92. We have consulted five relevant literature to illustrate that the SPS technology inherent features such as rapid heating rates, short sintering times, and applied external pressure (which improves the densification), enable excellent sintering while avoiding formation of microstructural issues related to e.g. excessive grain growth, development of porosity, or microstructural heterogeneities.
Thank you very much for your comments, which has greatly improved our paper from professional level to standard format.
Reviewer 2 Report
Authors made excellent job addressing all the reviewer and improving the discussion. However the paper is still weak. A secondary aim should be reflected on the CLINICAL application of the material studied and investigated. For this reason It should be strongly recommended to still increase the discussion section adding some references as suggested:
Cicciù, Marco et al. “Facial Bone Reconstruction Using both Marine or Non-Marine Bone Substitutes: Evaluation of Current Outcomes in a Systematic Literature Review” Marine drugs vol. 16,1 27. 13 Jan. 2018, doi:10.3390/md16010027
Nary Filho H., Pinto T.F., de Freitas C.P., Ribeiro-Junior P.D., dos Santos P.L., Matsumoto M.A. Autogenous bone grafts contamination after exposure to the oral cavity. J. Craniofac. Surg. 2014;25:412–414. doi: 10.1097/SCS.0000000000000682
Figueiredo A., Coimbra P., Cabrita A., Guerra F., Figueiredo M. Comparison of a xenogeneic and an alloplastic material used in dental implants in terms of physico-chemical characteristics and in vivo inflammatory response. Mater. Sci. Eng. C Mater. Biol. Appl. 2013;33:3506–3513.
Author Response
Dear Reviewer:
First of all, thank you very much for your interest and recognition in our research work, which has greatly enhanced our confidence. We have carefully studied your valuable opinions on us and have revised and supplemented them one by one. Revised portion are marked in blue in the paper. The responds are as following:
Point 1: A secondary aim should be reflected on the CLINICAL application of the material studied and investigated. For this reason It should be strongly recommended to still increase the discussion section adding some references as suggested:
I. Cicciù, Marco et al. “Facial Bone Reconstruction Using both Marine or Non-Marine Bone Substitutes: Evaluation of Current Outcomes in a Systematic Literature Review” Marine drugs vol. 16,1 27. 13 Jan. 2018, doi:10.3390/md16010027
II. Nary Filho H., Pinto T.F., de Freitas C.P., Ribeiro-Junior P.D., dos Santos P.L., Matsumoto M.A. Autogenous bone grafts contamination after exposure to the oral cavity. J. Craniofac. Surg. 2014;25:412–414. doi: 10.1097/SCS.0000000000000682
III. Figueiredo A., Coimbra P., Cabrita A., Guerra F., Figueiredo M. Comparison of a xenogeneic and an alloplastic material used in dental implants in terms of physico-chemical characteristics and in vivo inflammatory response. Mater. Sci. Eng. C Mater. Biol. Appl. 2013;33:3506–3513.
Response 1: Considering the Reviewer’s suggestion, we have added discussion in several parts of the paper.
In lines 69-74, we added the information about how HAP composites can reduce the incidence of inflammation when they are transferred to fill bone defects.
In lines 69-74, we quoted articles about the bacterial count of Staphylococcus on the surface of dense HAP composites to a low level. The composites have advantages in avoiding infection and complications when it is used as an intraosseous implant to repair alveolar ridge after tooth loss.
In lines 251-255, we discussed the advantages of synthetic biomaterials over autogenous bone.
Finally, in lines 251-255, we discussed the possible clinical application of the material, such as scaffold materials, bone grafts used in dentistry, filling bone defects and facial bone reconstruction for bone regeneration/repair.
Including several references recommended by reviewers, 18 references were added to the revision.
Thank you very much for your comments, which has greatly improved our paper from professional level to standard format.